# Study on the Relationship between Emulsion Properties and Interfacial Rheology of Sugar Beet Pectin Modified by Different Enzymes

**DOI:** 10.3390/molecules26092829

**Published:** 2021-05-10

**Authors:** Yongjie Zhou, Yuqi Mei, Tian Luo, Wenxue Chen, Qiuping Zhong, Haiming Chen, Weijun Chen

**Affiliations:** 1College of Food Sciences & Engineering, Hainan University, 58 People Road, Haikou 570228, China; 18085231210032@hainanu.edu.cn (Y.Z.); 20170881310017@hainanu.edu.cn (Y.M.); 20180881310069@hainanu.edu.cn (T.L.); chwx@hainanu.edu.cn (W.C.); 990511@hainanu.edu.cn (Q.Z.); 2Huachuang Institute of Areca Research-Hainan, 88 People Road, Haikou 570208, China; 3Chunguang Agro-Product Processing Institute, Wenchang 571333, China

**Keywords:** sugar beet pectin, enzymatic modification, interfacial rheology, adsorption kinetics, emulsifying property

## Abstract

The contribution of rheological properties and viscoelasticity of the interfacial adsorbed layer to the emulsification mechanism of enzymatic modified sugar beet pectin (SBP) was studied. The component content of each enzymatic modified pectin was lower than that of untreated SBP. Protein and ferulic acid decreased from 5.52% and 1.08% to 0.54% and 0.13%, respectively, resulting in a decrease in thermal stability, apparent viscosity, and molecular weight (Mw). The dynamic interfacial rheological properties showed that the interfacial pressure and modulus (E) decreased significantly with the decrease of functional groups (especially proteins), which also led to the bimodal distribution of particle size. These results indicated that the superior emulsification property of SBP is mainly determined by proteins, followed by ferulic acid, and the existence of other functional groups also promotes the emulsification property of SBP.

## 1. Introduction

Commercial pectins are extracted from citrus peel and apple pomace in most instances [1]. A relatively new type of pectin, sugar beet pectin (SBP), has recently received much attention because of its excellent emulsifying ability [2,3]. In addition, sugar beet pulp is generally discarded as a by-product of the sugar industry, and SBP is usually extracted from sugar beet pulp. However, the poor gelling properties and thickening stability of SBP have limited its industrial production [4]. Generally, SBP consists of linear chains of α-1,4-linked galacturonic acid (GalA) units interrupted by the insertion of (1-2)-linked L-rhamnopyranosyl residues [5,6]. The chains also have branches, which means some rhamnosyl residues were substituted by arabinose, galactose, rhamnose and 13 other monosaccharides. In addition, lateral chains contained phenolic acids such as ferulic acid (FA), which was linked to the arabinose and galactose residues via ester linkages [7]. Furthermore, a relative higher concentration of proteinaceous materials were bounded to the side chains through covalent linkages [8]. The GalA unit of SBP was partially methyl esterified at C6 (methylation) or O-acetylated at C2 and/or C3 (acetylation), which was an important indicator for evaluating the functional properties and sources of SBP [9]. As mentioned earlier, SBP does not form gels like traditional pectin due to its special structural properties, but it has excellent emulsifying ability [10]. In recent decades, researchers have studied the relationship between SBP structure and emulsifying ability by modifying SBP structure. Some scholars suggested that the emulsifying ability of SBP can be explained by the high percentage content of acetyl groups [11]. However, some scholars have also studied the relationship between its chemical structure and emulsifying ability. There is no evidence that its emulsifying ability is related to the number of acetyl groups, but it is related to the high protein content [12]. The results showed that the emulsification performance of deproteinized SBP (protein content decreased from 5% to 0.5%) was worse than that of natural SBP [13]. Siew et al. pointed out that SBP with a higher protein content (12%) was more likely to adsorb to oil droplets and reduce the oil-water interfacial tension, further confirming the key role of protein [14]. In addition, as reported by Saulnier FA esters are widely distributed in the side chain of SBP, which greatly affects the emulsifying performance of SBP [3]. Leroux et al. suggested that high methylated pectin (HMP) could reduce the interfacial tension between the water and oil phases, possibly due to the action of its hydrophobic groups, such as the COOCH_3_-group. [12]. There are few reports on the effect of arabinose and galactose on the emulsifying ability of pectin.

The purpose of this study was to investigate the relationship between emulsifying ability and structural modification of SBP. In detail, the samples were prepared by modifying SBP with protease (PE), endo-α-1,4-polygalacturonase (PG), endo-β-1,5-arabinanase (ABN), endo-β-1,4-galactanase (GAL), feruloyl esterase (FAE), pectin methyl esterase (PME). The structures of SBP were characterized by NMR and FT-IR. Moreover, the dilatational viscoelastic properties of pectin solutions at the oil-water interface were also measured to reveal the stabilized mechanism for emulsions. The study of the adsorption behavior of pectin and the interfacial rheological properties of the adsorbed layer can provide some useful insights into the mechanical causes of the emulsion [15].

## 2. Results and Discussion

### 2.1. Effect of Enzymatic Modification on SBP Components

Different enzymatic modifications were used to modify SBP, and the composition of pectin samples obtained was shown in Table 1. Each enzyme treatment effectively degraded the corresponding target content. The concentration of protein decreased from 5.52% to 0.54% when the enzyme degradation time was 16 h by using PE as a proteinaceous decomposing material, and these values were almost equivalent to those reported previously [13]. The total protein contents of SBP_ABN_, SBP_GAL_ and SBP_ABN+GAL_ were decreased from 5.52% to 4.87%, 4.94% and 2.22%, respectively. The possible reason is that arabinose and galactose are mainly distributed in the side chain, and proteins attach preferentially to these two monosaccharides in the side chain rather than the main chain, and this is also the case of FA [7]. The decrease of protein concentration in SBP_FAE_ may be due to the cross-linking of protein and FA [16]. In addition, the protein contents of all samples were in the range of 0.54–5.52%, which falls into the previously reported range of 0.1–6.5% [12,17].

FA is a minor phenolic moiety present in the side chain of SBP. According to Table 1, the total FA contents of pectin samples treated with FAE, ABN, GAL and ABN+GAL was remarkably reduced. The order of decreasing the contents was SBP_FAE_ (0.13%) < SBP_ABN+GAL_ (0.52%) < SBP_GAL_ (0.74%) < SBP_ABN_ (0.87%). The reason for this phenomenon is consistent with the above explanation of the decrease in protein content.

The effect of PG mainly occurs in the backbone of SBP because it is used to randomly hydrolyze the connections between GalA [18]. In our study, the content of GalA in SBP_PG_ was 32.03%, while other enzymatic modifications did not significantly change it.

The DE expressed as the number of methylated GalA per 100 GalA units [7]. PME was used to hydrolyze methyl esters of pectin to pectate and methanol. Therefore, the DE values of SBP_PME_ decreased from 85.48% to 8.41%, while the concentrations of other components changed slightly. Moreover, the DE values of all samples ranged from 82.85–85.48% (Table 1), belonging to high methoxy pectin, which was also confirmed by subsequent FT-IR and ^1^H-NMR spectroscopy.

### 2.2. Molecular Weight Distribution Analysis

Molecular weight (Mw) distribution was generally considered as an important parameter, which was highly related to the physicochemical properties of pectin. Furthermore, the Mw of pectins depends on its main chain and side chains [19]. The Mw data of each pectin sample was shown in Table 2. The Mw of pectin treated with each enzyme was lower than that without treatment. The decrease in the Mw was greater in the order: SBP_PE_ (184.8 kDa) < SBP_ABN+GAL_ (207.9 kDa) < SBP_FAE_ (221.5 kDa) < SBP_ABN_ (233.5 kDa) < SBP_GAL_ (241.8 kDa) < SBP_PG_ (258.1 kDa). The reasons for the decrease in Mw mainly related to the loss of protein, especially the protein bound to the side chain [15]. In addition, the ratio of Mw/Mn can indicate the different distribution of pectin. The larger the ratio of Mw/Mn, the wider the molar mass distribution, and the smaller the ratio, the narrower the molar mass distribution [18]. In our results, the Mw/Mn of all samples ranged from 2.37 to 2.93, which showed that the distribution of molecular weight was relative narrow. These values showed that SBP were highly homogeneous polysaccharides with concentrated molecular-weight distributions [20,21].

### 2.3. Fourier Infrared Spectroscopy Analysis (FT-IR) of Samples

The FT-IR analysis of pectin samples is shown in Figure 1. The strong signals in the range of 1700–1750 cm^−1^ and 1600–1650 cm^−1^ are related to free carboxyl groups and esterification in GalA, respectively. The absorption in the range of 1000~1200 cm^−1^ was mainly from the C-O of glycosides [22,23]. Therefore, there was no obvious difference in the spectra of SBP, SBP_PE_, SBP_ABN_, SBP_GAL_ and SBP_ABN+GAL_. FAE mainly degrades ferulic acid groups attached to arabinose and galactose through esterification reaction, resulting in a large number of ester bond breakage at the end of the reaction, which is also the reason why the spectra of SBP_FAE_ vary greatly (especially –COOCH_3_) in the range of 1700–1750 cm^−1^ and 1600–1650 cm^−1^ [23]. In addition, the spectrum of SBP_PME_ was similar to that of SBP_FAE_, because PME is mainly used to remove the methoxy residues in pectin to generate polygalacturonic acid. Compared with SBP, the signal intensity of COOH of SBP_PG_ during 1600–1650 was obviously weakened because PG hydrolyzed GalA. Also, the high intensity of characteristic peak at 1700–1750 cm^−1^ represents high DE value (higher than 50%) for all pectin samples [24], which was in line with the results of DE measurement.

### 2.4. Nuclear Magnetic Resonance (NMR) Analysis

The modification process of SBP was analyzed by NMR spectroscopy (Figure 2). The signal peak at δ = 1.1 ppm represents the fatty acid hydrogen on the ferulic acid molecule. Therefore, compared with other samples, SBP_FAE_, SBP_ABN+GAL_ and SBP_PE_ (especially SBP_FAE_) have significantly lower signal peak values at δ = 1.1 ppm. In addition, the peaks at δ = 1.18 ppm represents the Rhap residues of methyl rhamnoses [25], and the δ = 1.2 ppm signal represents hydrogen on O-CH_3_, which is one of the reasons for the peaks decreasing of SBP_ABN_ and SBP_GAl_. The intensity of the peaks at δ = 1.27 was corresponded to the CH_2_ groups in the anhydride molecule. The signal at around δ = 1.99 ppm was attributed to the acetyl groups binding at GalA residues in SBP, which was notably different from commercial citrus pectin [26]. Furthermore, the intensities of the peaks at δ = 1.93, 1.85 and 2.12 ppm, which were indicative of fatty-acid chains, and the signal near δ = 2 ppm represented hydrogen on CH_3_ on the acetyl group. It can be seen from Figure 2 that the signal peak intensity of each sample was not notably different at about δ = 2 ppm, indicating that neither enzymatic modification occurs at the fatty chain and the methyl group of an acetyl group.

### 2.5. Thermal Analysis

The thermal properties of SBP, especially its thermal stability, affect its range of applications, such as high temperature sterilization. Thermogravimetric analysis (TG) was used to analyze the thermal characteristics of SBP to further investigate the structure-function relationship. As shown in Figure 3, all samples had a continuous weight loss throughout the temperature range and were divided into three stages. As the temperature increased, a slight mass loss was happened in the first stage (from room temperature to about 150 °C), which was caused by the evaporation of water in the sample. In the water loss process, the amount of loss was similar for each sample. In the second stage (About 200 °C to 300 °C), the carbonaceous residue decomposition of samples led to a rapid loss of mass (approximately 50%). There were obvious differences in thermogravimetric behavior. Although the mass of each sample drops sharply between 200 and 300 temperatures, SBP was the first to reach equilibrium and then slowly declines, while SBP_PE_ was the last to reach equilibrium. The third stage (300–550 °C) showed the slow mass loss may be due to the thermal decomposition of solid char. As the temperature increased, solid carbons containing polyaromatic hydrocarbon structures grafted with aliphatic and ketone groups will partially damage and piled up tightly [27]. All of them decomposed in a wide temperature range (200–550 °C), with a final residue yield of 29.15%, 28.96%, 27.53%, 27.24%, 26.55%, 25.91%, 24.23%, and 20.40% for SBP, SBP_PME_, SBP_PG_, SBP_GAL_, SBP_ABN_, SBP_FAE_, SBP_ABN+GAL_, and SBP_PE_ respectively. Therefore, it can be concluded that SBP with higher protein content exhibited good thermal stability.

### 2.6. Scanning Electron Microscope (SEM)

Appreciable differences were observed in the surface topographies of the pectin samples when the samples were imaged using SEM (Figure 4). The original pectin (SBP) had a relatively isolated, smooth and dense surface, SBP_PME_ and SBP_PG_ had a rougher, coral-like structure, and SBP_PE_, SBP_FAE_, SBP_ABN_, SBP_GAL_ and SBP_ABN+GAL_ had a filamentous structure. It was reported that the surface topography of pectin samples depended on their Mw [28,29]. In general, it is precisely because enzyme digestion reduces the molecular weight of pectin, the easier it is to form irregular structures, which was consistent with the above results of Mw.

### 2.7. Adsorption Kinetics at the Oil-Water Interface

The change of the interfacial pressure with the adsorption time was shown in Figure 5A. The interfacial pressure of all samples increased with the prolongation of time, which was attributed to the samples in the solution diffusing to the interface and adsorbing on it. The curve of samples did not reach adsorption equilibrium until 7200 s because the adsorption course of macromolecular surfactants typically requires 2–3 days to reach equilibrium [30]. The curve for SBP_PME_ and SBP_PG_ were similar to that of SBP, indicating that the decrease of esterification degree and galacturonic acid content did not affect the interface behavior of SBP. However, the interfacial pressure of SBP_ABN+GAL_, SBP_FAE_ and SBP_PE_ increased tardily to a lower value than all other samples, suggesting that these three samples had weaker surface activity, which was due to the significant decrease of protein and ferulic acid content.

There was a rate-determining step, displaying relatively low interfacial pressures during the first step [31]. The relationship between interfacial pressure (*π*) and adsorption time (*t*) can be associated with a modified form of the Ward-Tordai equation [32]:(1)$$\pi \left(t\right)=2C_{0}KT{\left(\frac{Dt}{3.14}\right)}^{1/2}$$
where *C*_0_ is the concentration in the bulk phase, *K* is the Boltzmann constant, *T* is the absolute temperature, and *D* is the diffusion coefficient. Diffusion is mainly driven by the concentration gradient. If protein diffusion controls the adsorption process, a plot of *π* versus *t*^1/2^ will be linear, and the diffusion rate (*K*_diff_) will be represented by the slope of this plot [33]. When the adsorption time exceeded 400 s, the *π*–*t*^1/2^ curve deviated completely from the straight line, and the adsorption kinetics was no longer controlled by diffusion but dominated by adsorption and rearrangement (Figure 5B,C).

To detect the dynamics of the unfolding and rearrangement of pectin molecules at the interface, the first-rate equation was used for analysis [34]:(2)$$\ln\left(\frac{\pi_{7200}-\pi_{t}}{\pi_{7200}-\pi_{0}}\right)=-k_{i}t$$
where *π*_7200_, *π*_0_, and *π*_t_ are the interfacial pressures at the *t* = 7200 s, at the *t* = 0 s, and at any time of each stage, respectively, and *k*_i_ is the first-order rate constant. As shown in Figure 5C, the equation produces two linear regions, the slope of the first linear region corresponds to the first-rate constant (*K*_p_) of pectin molecules unfolding at the oil-water interface, and the slope of the second linear region corresponds to the first-rate constant (*K*_r_) of pectin molecules rearranging at the oil-water interface [35].

As shown in Table 3, the rate constants of diffusion (*K*_diff_), penetration (*K*_p_), reorganization (*K*_r_), and the interfacial pressure (*π*_7200_) of pectin samples were calculated and summarized. Significant differences were found in the equilibrium interfacial pressure of each pectin sample at the end of the experiment. SBP had the highest value of 11.22 mN/m, while SBP_PE_ was only 2.81 mN/m. There, it also means that proteins play a decisive role in the interface behavior, which can be proved by the lower interfacial pressure of SBP_ABN+GAL_ (4.89 mN/m) and SBP_FAE_ (5.35 mN/m).

In addition, from the data of *K*_diff_, *K*_p_ and *K*_r_, the diffusion, adsorption and recombination rates of pectin molecules at the oil-water interface are consistent with the results of interface pressure. The rate of each step decreased as the protein and ferulic acid content decreases. Moreover, the linear regression coefficients (LR) of each rate (*K*_diff_, *K*_p_ and *K*_r_) were greater than 0.9000, indicating the high reliability of these data. These results showed that the surface activity, interfacial pressure and interfacial behavior of pectin decreased with the decrease of protein and ferulic acid content in pectin molecules.

### 2.8. Dilatational Rheological Properties at the Oil-Water Interface

The dilatational viscoelastic properties reflect the adsorption behavior of molecules at the oil-water interface and predict the stability of emulsions as well as the interaction between molecules [16,36]. The interface dilatational modulus (E) which is caused by small changes of the surface area (dilatational strain) and the interface tension (γ) (dilatational stress), reflects the mechanical strength of the interfacial layer of the conjugate [37]. As shown in Figure 6, the E values of all samples increased over time, indicating that the mechanical strength of the oil-water interface improved with the increase of absorbing amounts of pectin samples. Of course, the E value of SBP was the largest from the beginning to the end, while the E value of SBP_PE_ only rises a little and finally approaches equilibrium and the lowest. These results were consistent with the results of interfacial pressure and adsorption kinetics. The dilatational elastic modulus (Ed) of interfacial layers formed by the pectin samples were presented in Figure 6B. There are some data points with negative values. As there is no negative elasticity this can only be a measurement artefact. Ed also increased gradually over adsorption time. After 7200 s of adsorption, the E_d_ value was equal to the dilatational modulus E (Figure 6A,B), while the values of the dilatational viscosity E_v_ were relatively small. Therefore, the adsorption process was mainly dominated by elastic behavior. In addition, E_d_ values of SBP_FAE_, SBP_ABN+GAL_ and SBP_PE_ changed in a small range at 7200 s, indicating that the degradation of protein and ferulic acid had great effect on the E_d_ of SBP. The Ed of SBP_PE_ was the lowest, suggesting that SBP_PE_ exhibits low elasticity at the interface.

As shown in Figure 6C, the dilatational modulus (E) versus interface pressure (*π*) was used to evaluate the amount of adsorption of samples [36]. The slop of the E-π curve could reflect the equilibrium state of pectins at the oil-water interface. E values of all samples increased immediately with the increase of *π* value, suggesting that the pectins adsorption at the interface increased. According to the theory of Lucassen-Reynders et al. [38], the interaction among the residues of the spread-out protein molecules increased with the increase of the protein adsorbed on the interface. In addition, the slopes (E-*π*) of all samples were greater than 1.0 and in a non-ideal adsorption behavior, suggesting that the systems were all dominated by intermolecular forces [39]. The slope of SBP (6.96) was higher than that of all samples, indicating the optimal interface behavior. With the decrease of the protein and ferulic acid proportion, the slope decreased significantly to 0.48 (SBP_FAE_), 0.41 (SBP_ABN+GAL_) and 0.16 (SBP_PE_), respectively. The slope of other samples also decreased with different enzymatic modification treatments (Figure 6C), the root cause is the reduction of active substances (especially protein and ferulic acid). These results suggested that the adsorption amounts of proteins at the oil-water interface reduced and interaction among molecules weakened.

### 2.9. Apparent Viscosity

The flow curves between apparent viscosity and shear stress of the sample solutions were showed in Figure 7. The apparent viscosity of all the dispersions decreased with the increase of shear rate (0.01–100 s^−1^), exhibiting typical shear-thinning behavior of pseudo-plastic fluids, indicating that the pectin solutions in this experiment were non-Newtonian fluids. Similar behaviors were observed in pectins from apple pomace [40], sugar beet pulp [41] and grapefruit peel [42]. This was because the increasing shear rate had promoted the fluidity of the pectin solution, and therefore, decreased the inter-molecular entanglement density between pectin molecules, resulting in a decrease in the viscosity of pectin solution. [43]. Therefore, SBP_PE_ has the lowest apparent viscosity of all samples because of the significant reduction in protein content, followed by SBP_ABN+GAL_. The pectin solution of SBP showed relative higher viscosity at the beginning than other samples, which was mainly related to the maximum Mw [44]. Schmelter et al. found the fracture of main chain and the decrease of side chain would lead to a lower Mw and viscosity [45]. The viscosity of SBP_PME_, SBP_PG_, SBP_ABN_, SBP_GAL_ and SBP_FAE_ also decreased, indicating that GalA, FA and RG-I also had an effect on the viscosity of the sample, and the order of the effect was FA > GalA > arabinose/galactose.

### 2.10. Emulsifying Properties

The influence of structure modification on emulsifying property was studied by measuring the particle size distribution and zeta potential. As shown in Figure 8A, both SBP_PE_ and SBP_ABN+GAL_ showed a bimodal distribution due to the significant reduction in protein content. SBP_FAE_ showed a wide unimodal distribution due to the decrease of ferulic acid group content on its side chain, while the particle size distribution of other pectin samples was also wider than that of SBP. These phenomena may be due to the less active molecules (protein, ferulic acid) in the pectin structure, which leads to the decrease of the adsorption amount at the oil-water interface, the decrease of the adsorption speed and the weakening of the interfacial film strength [46]. Also, the polydispersity index (PdI) value provides an important parameter for better interpretation of emulsion particle size. As can be seen from Table 4, all the PdI values of samples stabilized emulsions are less than 0.3, indicating that all the emulsion samples are uniform [21].

Since electrostatic repulsion was one of several factors that contributing to the stability of the emulsion to prevent coalescence, the zeta-potential was measured to assess the effect of static electricity on the stability of the emulsion (Figure 8B). All the pectin-stabilized emulsions presented negative charges. The charge of emulsions adding SBP_PE_ (−52.08 mV), SBP_FAE_ (−49.07 mV) and SBP_ABN+GAL_ (−50.15 mV) increased significantly compared to the control (SBP) which was −47.41 mV, while decreased when the emulsions stabilized by SBP_PG_ (40.78 mV) and SBP_PME_ (−43.82 mV). The results suggested that the pectins adsorbed at the oil-water interface decreased or the rearrangement of pectins adsorbed at the oil-water interface was occurred [47]. Among them, the zeta-potential of SBP_PE_, SBP_FAE_ and SBP_ABN+GAL_ stabilized emulsion increased because of the decrease of protein content and the increase of spatial repulsion, which has been confirmed by particle size results.

## 3. Materials and Methods

### 3.1. Materials and Chemicals

SBP was kindly provided by CP Kelco (Lille Skensved, Denmark). Medium-chain triglyceride (MCT, naturally occurring in coconut oil) was purchased from the Nisshin Oillio Group (Tokyo, Japan). Pepsin (E.C. 3.4.23.1) and food grade acid protease were obtained from Aladdin reagents (Shanghai, China). Endo-polygalacturonase (PG, EC. 3.2.1.15), endo-α-1,4-arabinanase (ABN, EC. 3.2.1.99), endo-1,4-β-galactanase (GAL, EC. 3.2.1.89), feruloyl esterase (FAE, EC. 3.1.1.73) and pectin methyl esterase (PME, E.C. 3.1.1.11) were purchased from Megazyme International Ireland Ltd. (Bray, Ireland). All other chemicals and solvents were analytical grade unless otherwise stated.

### 3.2. Enzymatic Modification of SBP

In order to complete the enzymatic modification of SBP’s structure, we followed the method reported by Chen et al., (2015) [18]. Briefly, a quantitative amount of SBP was dissolved in 0.1 M citrate buffer (pH 3.0) to prepare a 2.0 *w*/*v*% pectin solution. The completely dissolved pectin solution was added with pepsin (700 U/g SBP) and food-grade acid protease (70 U/g SBP) and placed in a water bath at 40 °C for continuous stirring until the reaction time was 14 h. Then, the solution was heated for 1 min at 100 °C to terminate the enzymatic reaction and dialyzed against distilled water at 25 °C using a dialysis membrane with a 10 kDa molecular weight cut off, followed by lyophilizing. The freeze-dried samples (SBP_PE_) were stored in a desiccator at 25 °C until use. Furthermore, the flowchart of at which stage the modification would appear to generate the new product (modified pectin) is shown in Figure 9. The other enzymatically modified samples (SBP_PME_, SBP_PG_, SBP_ABN_, SBP_GAL_, SBP_ABN+GAL_ and SBP_FAE_) were prepared according to the method described by Chen et al. [18] and above.

### 3.3. Determination of SBP’s Composition after Modification

#### 3.3.1. Protein

The protein content of pectin samples was determined by Coomassie Brilliant Blue reagent based on their absorbance at 595 nm. BSA was used as a standard for calibration [48].

#### 3.3.2. Ferulic Acid (FA)

The total content of FA in pectin samples were determined according to [49] by ultraviolet spectrophotometry. Briefly, 20 μg/mL FA standard solution was prepared by PBS (0.005 M, pH = 7.0). The absorbance of 0.1 *w*/*v*% pectin solutions were measured at 325 nm employing FA as standard.

#### 3.3.3. Galacturonic Acid (GalA)

The content of GalA in the pectin samples were performed by the m-hydroxyphenol method [50]. The standard of GalA (0.007%, *w*/*v*), 0.0125 M borax (sodium tetraborate) sulfuric acid solution and 1.5 mg/mL hydroxybiphenyl color developer were prepared respectively. Then, the absorbance of 0.1 *w*/*v*% pectin solutions were measured at 520 nm employing GalA as standard.

#### 3.3.4. Degree of Esterification (DE)

The DE of pectin samples was obtained according to the previously reported literature [51]. Briefly, 0.1 g of the pectin sample was soaked with 1 mL anhydrous ethanol, then 1 g of sodium chloride was added and completely dissolved in 100 mL of ultra-pure water. Phenolphthalein was used as an indicator, the solution was titrated with 0.1 M NaOH until the color was just pink and did not fade in 30 s, and the volume of NaOH consumed was recorded as V1. After that, 15 mL NaOH (0.25 M) was added to the pectin solution and stirred continuously for 30 min at room temperature. Then 15 mL HCl (0.25 M) was added and titrated with 0.1 M NaOH until it just turned pink and stayed for 30 s. The volume of NaOH consumed at this time was denoted as V2. The formula used to calculate the degree of esterification (DE) was the following:(3)$$\mathrm{DE}\left(\%\right)=\frac{V_{2}}{V_{1}+V_{2}}\times 100$$

#### 3.3.5. Molecular Weight Distribution

The molecular weight distribution of the pectin samples was determined by Gel permeation chromatography coupled with multi-angle laser light scattering (GPC-MALLS, 1260, Agilent Technologies Inc. Santa Clara, CA, USA). The GPC-MALLS system consisted of a PL Aquagel-OH MIXED-H pump (Agilent Technologies Inc.). The eluent (Millipore water) was delivered at a flow rate of 0.5 mL/min. The pectin sample solution (1 mg/mL) was filtered by a 0.45 μm filter membrane before testing on the machine. The value of d*_n_*/d*_c_* used for analyzing with ASTRA software was 0.131 mL/g (Version 5.3.4.14, Wyatt Technology, Santa Barbara, CA, USA).

### 3.4. Fourier Transform Infrared (FT-IR) Spectroscopy

FT-IR analysis of the pectin samples (blended with KBr powder in a weight ratio of 1:100, *w/w*) were determined using a FT-IR spectrometer (TENSOR 27, Bruker Optics, Ettlingen, Germany) at room temperature. The scanning was conducted in the absorbance mode at a resolution of 4 cm^−1^ within the frequency range 4000–400 cm^−1^. The absorption spectrum was obtained after denoising and baseline correction.

### 3.5. Nuclear Magnetic Resonance (NMR) Spectroscopy Analysis

^1^H-NMR spectra of the pectin samples were recorded on a Unity Inova 500 MHz spectrometer (Varian, Palo Alto, CA, USA). The pectin sample (20 mg) was completely dissolved in 0.5 mL 99.9% D_2_O, and the spectrum was obtained by ^1^H-NMR scanning 256 times.

### 3.6. Thermal Analysis

The thermodynamic characteristics of pectin samples were analyzed by TG on a simultaneous thermal analyzer (Netzsch STA 449C, Aldridge, UK). Pectin samples (10 mg) were placed on an aluminum pan (sealed immediately) and heated from 60 °C to 550 °C at a rate of 10 °C/min [52].

### 3.7. Scanning Electron Microscopy (SEM)

The freeze-dried pectin powder was glued to the plate with double-sided adhesive and then sprayed with gold. The morphological characteristics of the samples were observed and photographed by scanning electron microscope (SEM, Supra 55, Zeiss, Oberkochen, Germany) with an accelerating voltage of 10 kV.

### 3.8. Measurement of Interfacial Pressure (π)

The interfacial pressure (*π*) of pectin samples at the oil-water interface were determined on an optical contact anglemeter (OCA25, Dataphysics Instruments Co., Ltd., Filderstadt, Germany) at 25 °C as described by Xiong et al. [53]. Each sample (1.0 *w*/*v*%) was dispersed in phosphate buffer saline (PBS, pH 7.4, 0.01 M) with stirring continuously until completely dissolved. Before measurement, each solution was placed in a syringe and equilibrated for 10 min to reach 25 °C. Subsequently, the aqueous phase (pectin solutions) and the oil phase (MCT) were placed in the syringe and an optical glass cuvette, respectively. Droplets of each sample (30 μL) were extruded in the optical glass cuvette and stood for 7200 s to ensure adsorption at the oil-water interface. The *π* values were calculated according to the Equation (4):(4)$$\pi \left(\mathrm{mN}/m\right)=\gamma_{0}-\gamma$$
where γ_0_ (mN/m) and γ (mN/m) are the interfacial tension of PBS (0.01 M, pH 7.4) and pectin solutions, respectively.

### 3.9. Measurement of Interfacial Viscoelasticity

The dynamic interfacial viscoelasticity of the pectin samples at the oil-water interface was investigated using an automated drop tensiometer (Tracker-H, Teclis, France) at 25 °C [53]. Each sample (1.0 *w*/*v*%) was dispersed in PBS (0.01 M, pH 7.4) with continuous stirring. Amplitude sweep (0.01–100% at 0.1 Hz) was applied to measure the linear viscoelastic regions. Periodic oscillation drops of the sample (30 μL) at a 10% amplitude (ΔA/A, within the linear regime) in a 0.1 Hz frequency were injected after equilibration for 60 s. The dilatational modulus (E) is a complex quantity consist of real (elasticity modulus, E_d_ = |E|cos δ) and imaginary parts (viscous modulus, E_v_ = |E|sin δ), where the phase angle (δ) between stress and strain represents the relative viscoelasticity of the interfacial absorbed layer. Tan δ is the loss angle tangent that represents the ratio of E_v_ and E_d_. The test time for each sample solution was 7200 s.

### 3.10. Rheological Properties

In order to determine the apparent viscosity of different pectin solutions, a Thermo HAAKE Rotation Rheometer (HAAKE MARS 40, Karl-sruhe, Germany) with a 35 mm parallel steel plate configuration was used. The flow behavior was determined as described by Khan, et al. [54]. Pectin solutions (3.0 *w*/*v*%) were prepared and the relationship between apparent viscosity and shear rate was evaluated. The shear rate was ranged from 0.1 to 100 s^−1^.

### 3.11. Evaluation of Emulsifying Properties

#### 3.11.1. Emulsion Preparation

Emulsion formula were composed of 1 *w*/*w*% pectin as emulsifier, 15 *w*/*w*% MCT as oil phase [18]. Each pectin sample (1 g) was dissolved in 80 mL ultra-pure water and stirred overnight until completely dissolved. Citric acid solution (10 *w*/*v*%) was used to adjust the final pH of the solution to 3.0. Ultra-pure water and 15 g of MCT were then added to achieve a mass of 100 g. The mixture was then pre-homogenized (24,000 rpm, 3 min) to form a coarse emulsion, which was homogenized three times under an ultra-high pressure homogenizer (IKA-ULTRA-TURRAX T25, IKA 190 Works, Inc., Wilmington, NC, USA) operating at 30 MPa to obtain the desired emulsions.

#### 3.11.2. Zeta-Potential and Droplet Size Distribution of Emulsions

The zeta-potential and droplet size distribution of the newly prepared emulsions at room temperature were measured by Laser particle size and zeta potential analyzer (Zetasizer Nano S90, Malvern Instruments, Malvern, UK) according to the method of Liang, et al. [55]. The emulsions were diluted 100 times with ultra-pure water before determination. All the measurements were performed in triplicate at 25 °C.

### 3.12. Statistical Analysis

All experiments were performed in triplicate unless otherwise stated. Analyses of variance were performed, and the mean values ± standard deviations were evaluated by Duncan’s multiple-range test (*p* < 0.05) using SPSS version 24.0 statistical software (SPSS Inc., Chicago, IL, USA). Origin (Origin Lab Co., Pro.9.0, Corporation, Northampton, MA, USA) software was used for data processing and to create charts.

## 4. Conclusions

The stability of SBP-stabilized emulsion after enzymatic modification, especially the contribution of the rheological properties and the viscoelasticity of the interfacial adsorbed layer to the stability of the emulsion were studied. The viscoelastic properties, thermal stability and apparent viscosity of the emulsion were reduced by enzymatic modification of pectin. On the other hand, the results of interfacial rheology showed that the reduction of protein and ferulic acid significantly reduced the diffusion, adsorption, and recombination rates of pectin at the oil-water interface, as well as the interfacial pressure at the end point of the adsorption. As a result, protein and FA played indispensable roles in emulsifying ability and stability, the decrease of methyl ester groups mainly affected the particle sizes of the emulsion. GalA, arabinose and galactose had less effect on emulsifying properties than other functional groups.

## Figures and Tables

**Figure 1 molecules-26-02829-f001:**
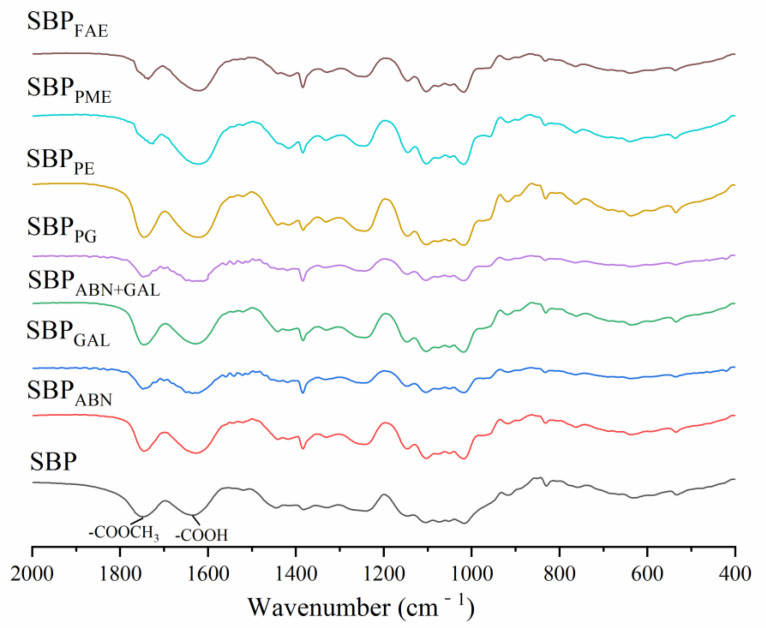
Fourier transform infrared spectra (FT-IR) of SBP after different modification methods.

**Figure 2 molecules-26-02829-f002:**
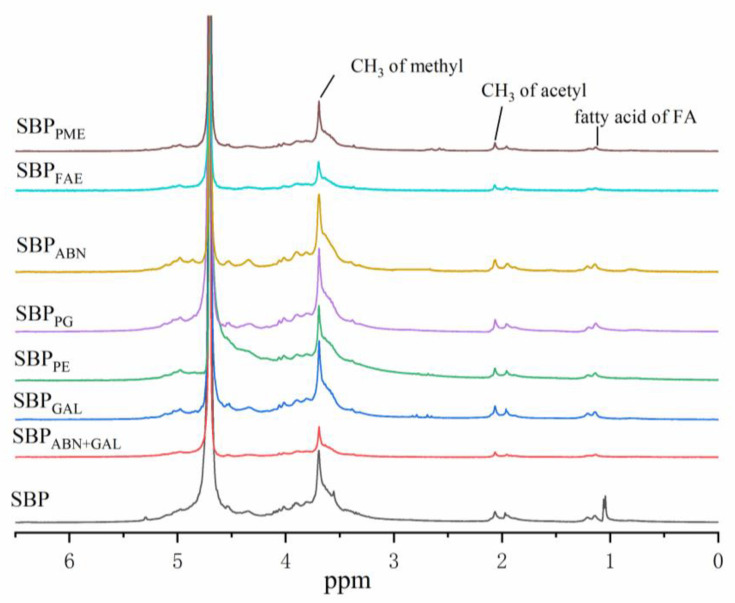
^1^H-NMR spectra of different SBP samples.

**Figure 3 molecules-26-02829-f003:**
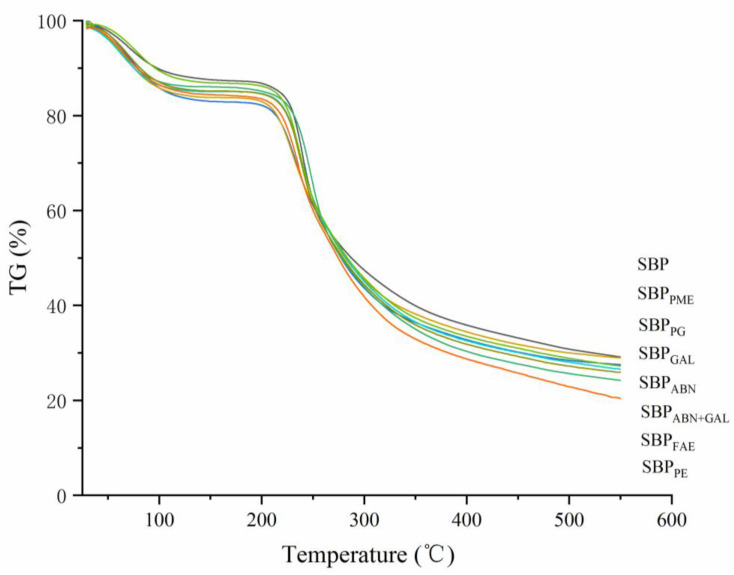
TG curves of different samples at a heating rate of 10 °C/min.

**Figure 4 molecules-26-02829-f004:**
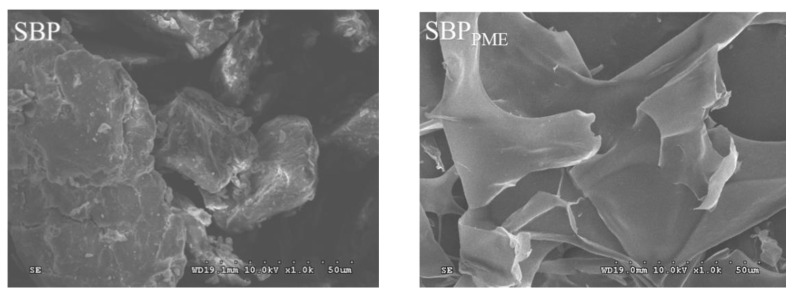
SEM for surface characteristics of different samples.

**Figure 5 molecules-26-02829-f005:**
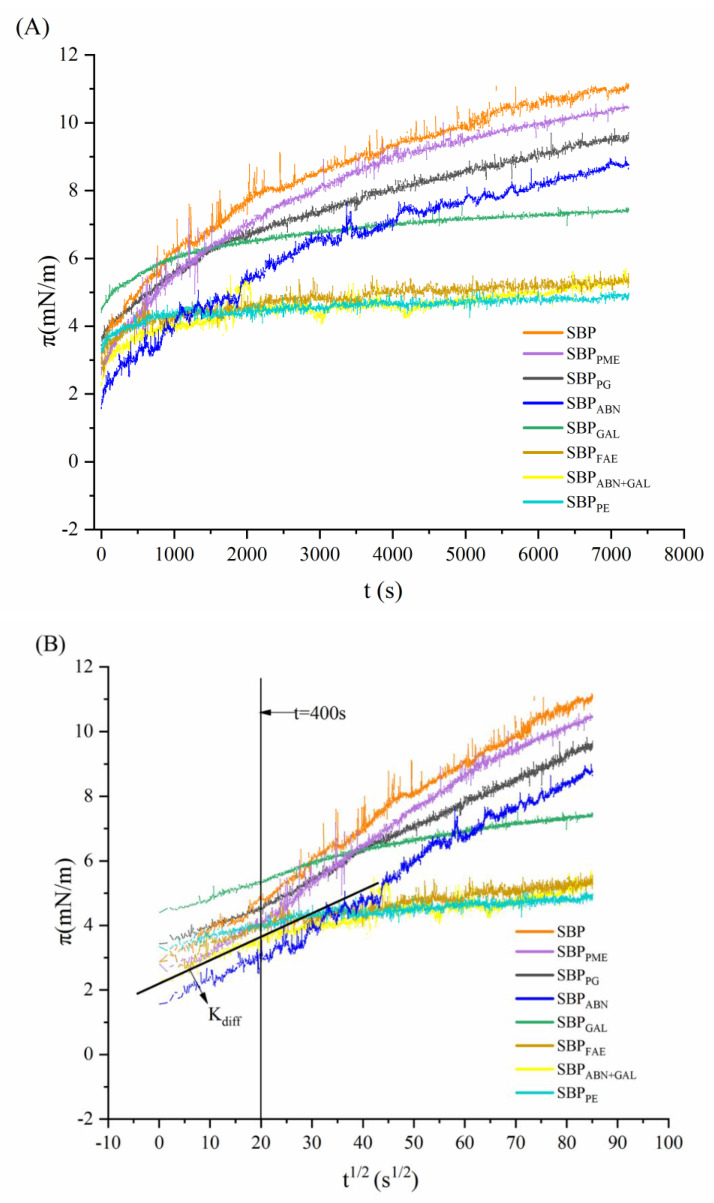
Dynamic interfacial pressure (*π*) for sample solutions at the oil-water interface (**A**). The *t*^1/2^-dependent π of sample solutions at the oil-water interface, *K*_diff_ represents diffusion rate (**B**). The profile of the molecular penetration and rearrangement steps at the oil-water interface for samples, *K*p and *K*r represent first-order rate constants of penetration and rearrangement (**C**), respectively.

**Figure 6 molecules-26-02829-f006:**
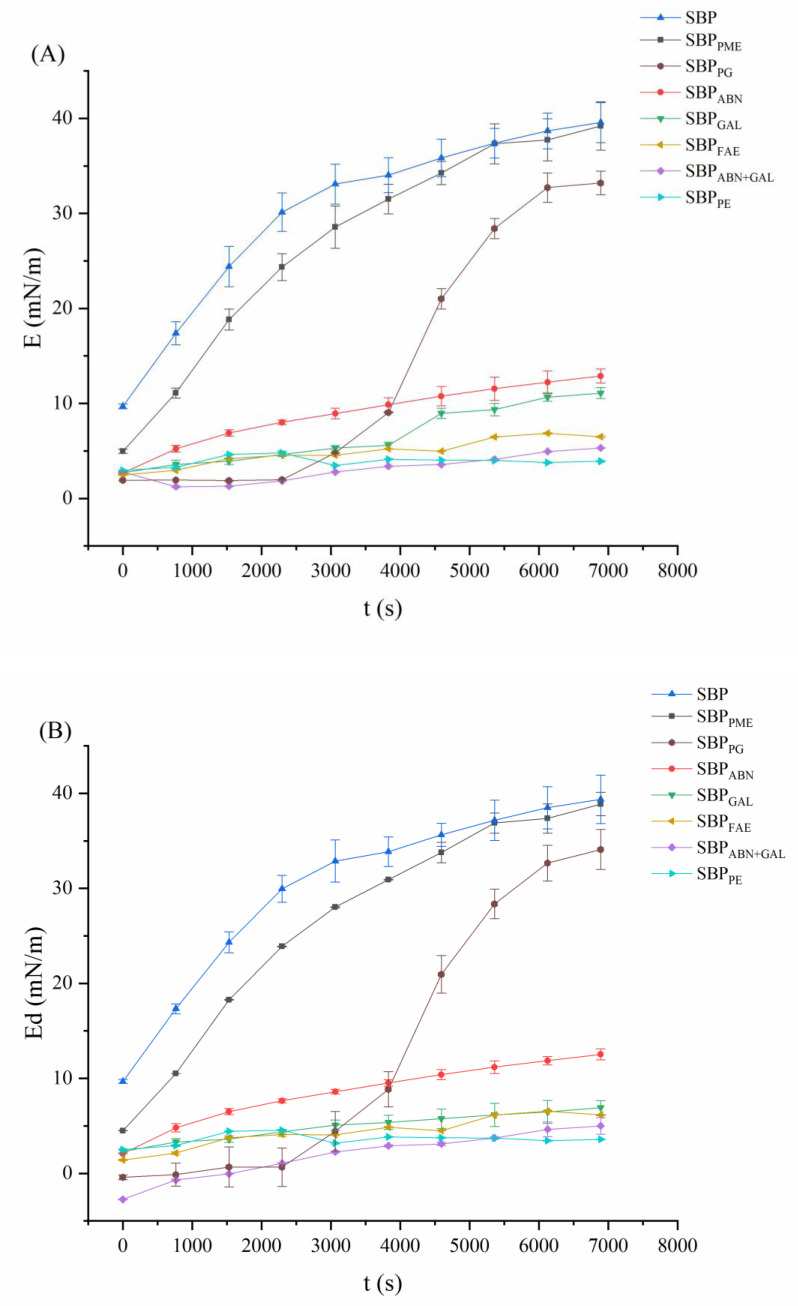
Time-dependent dilatational modulus (E) and dilatational elasticity (E_d_) for samples at the oil-water interface (**A**,**B**). Interface dilatational modulus (E) as a function of interface pressure (*π*) for samples at the oil-water interface (**C**).

**Figure 7 molecules-26-02829-f007:**
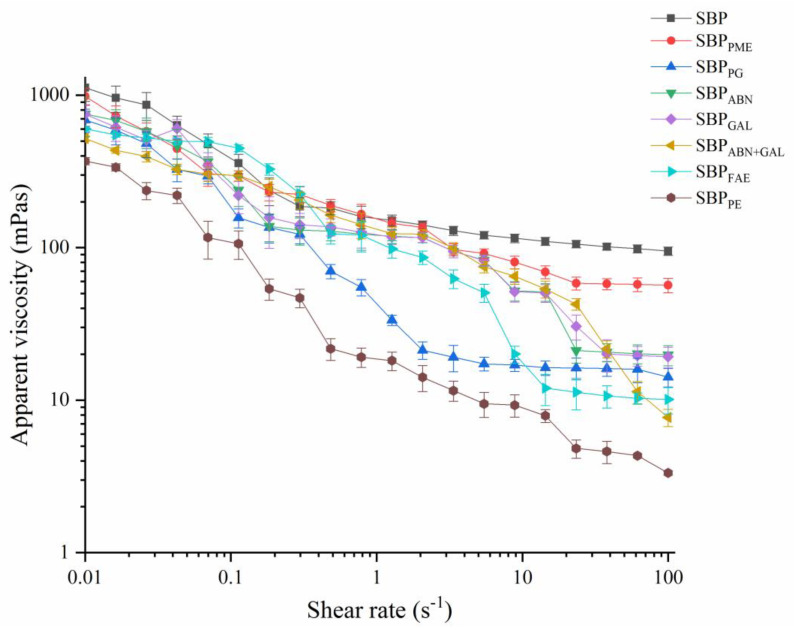
Flow curves of different sample solutions.

**Figure 8 molecules-26-02829-f008:**
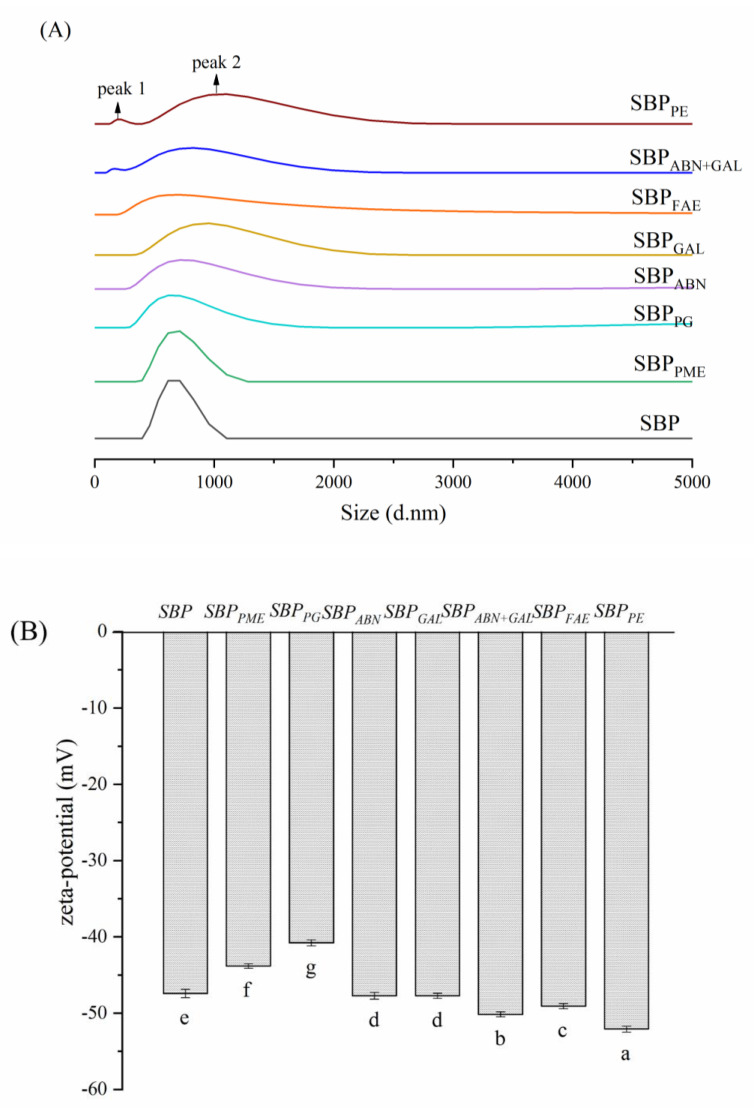
Particle size distribution (**A**) and zeta- potential (**B**) of emulsions stabilized by different samples.

**Figure 9 molecules-26-02829-f009:**
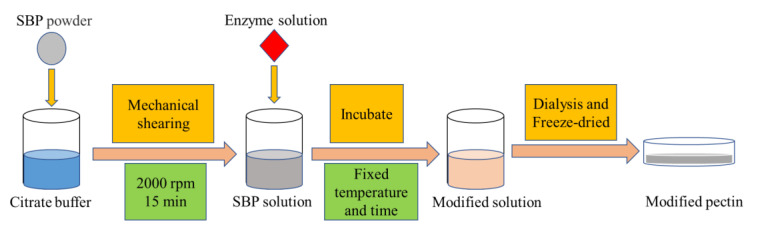
The flowchart of at which stage the modification would appear to generate the new product (modified pectin).

**Table 1 molecules-26-02829-t001:** The compositions of SBP after different modifications.

	Protein (%)	Ferulic Acid (%)	GalA (%)	DE (%)
SBP	5.52 ± 0.01 ^a^	1.08 ± 0.02 ^a^	65.88 ± 1.77 ^c^	85.48 ± 5.81 ^a^
SBP_ABN_	4.87 ± 0.02 ^e^	0.87 ± 0.02 ^e^	66.67 ± 1.70 ^b^	83.58 ± 2.02 ^e^
SBP_GAL_	4.94 ± 0.04 ^d^	0.74 ± 0.01 ^f^	63.02 ± 0.78 ^f^	84.38 ± 0.88 ^d^
SBP_ABN+GAL_	2.22 ± 0.01 ^g^	0.52 ± 0.05 ^g^	67.80 ± 0.95 ^a^	83.31 ± 2.70 ^f^
SBP_PG_	5.29 ± 0.03 ^c^	1.02 ± 0.01 ^c^	32.03 ± 2.10 ^h^	85.03 ± 1.44 ^b^
SBP_PE_	0.54 ± 0.01 ^h^	0.98 ± 0.01 ^d^	62.21 ± 0.75 ^g^	84.70 ± 1.10 ^c^
SBP_PME_	5.43 ± 0.03 ^b^	1.04 ± 0.01 ^b^	65.28 ± 1.53 ^d^	8.41 ± 1.67 ^h^
SBP_FAE_	4.32 ± 0.01 ^f^	0.13 ± 0.03 ^h^	63.45 ± 1.53 ^e^	82.85 ± 1.97 ^g^

The data are averages and standard deviations of triplicate measurements. Values in each column with different superscript letters (a–h) are significantly different (*p* < 0.05).

**Table 2 molecules-26-02829-t002:** The weight-average molecular weight (Mw), Mw/Mn of SBP after different modifications.

Samples	Mn (kDa)	Mw (kDa)	Mw/Mn
SBP	120.7 ± 10.47 ^a^	286.6±3.61 ^a^	2.37 ± 0.20 ^g^
SBP_ABN_	88.1 ± 3.07 ^d^	233.5 ± 4.56 ^e^	2.65 ± 0.08 ^d^
SBP_GAL_	82.58 ± 2.98 ^e^	241.8 ± 3.66 ^d^	2.93 ± 0.13 ^a^
SBP_ABN+GAL_	72.10 ± 3.30 ^g^	207.9 ± 2.31 ^g^	2.88 ± 0.29 ^b^
SBP_PG_	104.6 ± 5.25 ^c^	258.1 ± 2.45 ^c^	2.47 ± 0.09 ^f^
SBP_PE_	55.2 ± 4.45 ^h^	184.8 ± 5.15 ^h^	2.47 ± 0.25 ^f^
SBP_PME_	107.9 ± 2.35 ^b^	275.7 ± 4.90 ^b^	2.55 ± 0.19 ^e^
SBP_FAE_	80.6 ± 2.01 ^f^	221.5 ± 5.19 ^f^	2.75 ± 0.09 ^c^

Mw, weight average molecular weight. Mn, number average molecular weight. Mw/Mn, index of dispersion. The data are averages and standard deviations of triplicate measurements. Values in each column with different superscript letters (a–h) are significantly different (*p* < 0.05).

**Table 3 molecules-26-02829-t003:** Characteristic parameters, including the diffusion rate (*K*_diff_), constants of penetration and rearrangement at the interface (*K*_p_ and *K*_r_), and interface pressure at the end of adsorption (7200 s, *π*_10800_) for each pectin sample.

	*K*_diff_ (mN/m/s^1/2^) (LR)	*K*_p_ × 10^−4^ (LR)	*K*_r_ × 10^−4^ (LR)	*Π*_7200_ (mN/m)
SBP	0.100 ± 0.002 (0.9867) ^a^	−4.22 ± 0.003 (0.9003) ^a^	−14.22 ± 0.003 (0.9203) ^a^	11.22 ± 0.03 ^a^
SBP_PME_	0.090 ± 0.002 (0.9879) ^b^	−4.15 ± 0.001 (0.9780) ^b^	−14.15 ± 0.001 (0.9580) ^b^	10.69 ± 0.09 ^b^
SBP_PG_	0.076 ± 00.001 (0.9518) ^c^	−3.63 ± 0.001 (0.9645) ^c^	−11.63 ± 0.001 (0.9545) ^c^	9.70 ± 0.01 ^c^
SBP_ABN_	0.032 ± 0.001 (0.9518) ^d^	−3.62 ± 0.005 (0.9279) ^d^	−8.62 ± 0.001 (0.9759) ^d^	9.12 ± 0.02 ^d^
SBP_GAL_	0.024 ± 0.002 (0.9042) ^e^	−3.46 ± 0.001 (0.9765) ^e^	−7.46 ± 0.001 (0.9765) ^e^	7.59 ± 0.41 ^e^
SBP_FAE_	0.023 ± 0.001 (0.9023) ^f^	−3.07 ± 0.003 (0.9311) ^f^	−5.07 ± 0.003 (0.9241) ^f^	5.35 ± 0.02 ^f^
SBP_ABN+GAL_	0.023 ± 0.001 (0.9122) ^g^	−3.06 ± 0.008 (0.9859) ^g^	−4.06 ± 0.005 (0.9859) ^g^	4.89 ± 0.02 ^g^
SBP_PE_	0.013 ± 0.001 (0.9340) ^h^	−2.98 ± 0.005 (0.9088) ^h^	−3.98 ± 0.003 (0.9748) ^h^	2.81 ± 0.02 ^h^

Different letters within the same column are significant (*p* < 0.05) by Duncan’s multiple range test. LR is an abbreviation for linear regression coefficients.

**Table 4 molecules-26-02829-t004:** The droplet mean diameters and PdI of emulsions.

	Droplet Mean Diameters (nm)	PdI
SBP	655 ± 20.17 ^a^	0.16 ± 0.01 ^a^
SBP_PME_	729 ± 10.35 ^b^	0.18 ± 0.02 ^b^
SBP_PG_	866 ± 21.22 ^c^	0.21 ± 0.06 ^c^
SBP_ABN_	972 ± 22.78 ^d^	0.24 ± 0.03 ^d^
SBP_GAL_	1069 ± 31.24 ^e^	0.25 ± 0.03 ^e^
SBP_FAE_	1275 ± 28.33 ^f^	0.25 ± 0.02 ^e^
SBP_ABN+GAL_	1377 ± 35.85 ^g^	0.27 ± 0.13 ^f^
SBP_PE_	1520 ± 24.66 ^h^	0.28 ± 0.03 ^g^

The data are averages of three measurements with standard deviation. Data with different letters (a–h) in a same column were significantly different (*p* <0.05).

## Data Availability

All data is included in the article.

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
