# Peer review of "Study on the Relationship between Emulsion Properties and Interfacial Rheology of Sugar Beet Pectin Modified by Different Enzymes"

_molecules, 2021, doi:10.3390/molecules26092829_

Round 1

Reviewer 1 Report

Thank you for submitting the manuscript “Study on the Relationship between Emulsion Properties and Interfacial Rheology of Sugar Beet Pectin Modified by Different enzymes” to Molecules. The manuscript is well written and well organized. The subject is interesting and important for the Molecules audience. The authors researched the enzymatic modification of pectin extracted from sugar beet pectin. I suggest that the authors add in the introduction if the raw material used in the extraction of pectin is normally a by-product of the sugar industry or include a flowchart of at which stage the modification would appear to generate the new product (modified pectin). In some places, minimal typing errors should be corrected so I suggest re-reading the entire text to verify these problems.

Reviewer 2 Report

Comment are in attached document

Reviewer 3 Report

This work studies the influence on the rheological properties of modifying a type of pectin that is commonly used as an emulsifying agent. In this way, it may be of interest and progress in the development of new formulations of food grade. Both the abstract and the introduccion section are well written and provide sufficient information. Although the characterization of the different modified pectins is well carried out, the most interesting results correspond to the interfacial and rheological data.

In my opinion, I believe that the paper could be considered for publication on Molecules after a review of some aspects. In addition, I include some questions that I would like to have answered by the authors.

  • Please provide the standard deviations or error bars in figures 6 and 7
  • Although it is obvious that there are different behaviors and the analysis developed may be valid, the adjustment of the data to a straight line is too rigorous. Is it common to bring this type of analysis to data with similar behaviors? Wouldn't it be better to replace figure 6c with a table that also indicates the values of R2?
  • I do not intend to criticize an article already published as Khan et al., but I can say that the protocol used for flow curves in these samples has not been optimal. The dispersion of the results and the shape of the flow curves is very strange. For example, the shape of the curve for the SPB sample appears to indicate sample ejection at high speeds. However, what surprises me most is not that, but that curves have not even been adjusted to a mathematical model or it has been indicated that the behavior is pseudoplastic (non-Newtonian is very general). I therefore propose to improve this analysis.
  • If emulsions are analyzed in the study, droplet size distributions, not particles, should be discussed.
  • The range of the X axis in Figure 8a is too wide, I suggest lowering it to a maximum of 4000-5000 nm. Why aren't data from droplet mean diameters and polydispersities provided?
  • Why hasn't the physical stability of emulsions been analyzed?

Round 2

Reviewer 3 Report

Although some of the suggestions have not been taken in t account by the authors, I consider that they have justified their decisions. In addition, they have answered all the questions raised and the manuscript in its current version has sufficient quality for publication in Molecules.